# Improving quality and use of routine health information system data in low- and middle-income countries: A scoping review

**Seblewengel Lemma**[1]*, **Annika Janson**[1¤a], **Lars-Åke Persson**[1], **Deepthi Wickremasinghe**[2†], **Carina Källestål**[1¤b]

**1** Department of Disease control, London School of Hygiene and Tropical Medicine, based at the Ethiopian Public Health Institute, Addis Ababa, Ethiopia, **2** Department of Disease Control, London School of Hygiene & Tropical Medicine, London, United Kingdom

† Deceased.
¤a Current address: Department of Women's and Children's Health, Karolinska Institutet, Stockholm, Sweden
¤b Current address: Department of Women's and Children's Health, Uppsala University, Uppsala, Sweden
* lemmaseblewengel@gmail.com, Seblewengel.abreham@lshtm.ac.uk

## Abstract

### Background

A routine health information system is one of the essential components of a health system. Interventions to improve routine health information system data quality and use for decision-making in low- and middle-income countries differ in design, methods, and scope. There have been limited efforts to synthesise the knowledge across the currently available intervention studies. Thus, this scoping review synthesised published results from interventions that aimed at improving data quality and use in routine health information systems in low- and middle-income countries.

### Method

We included articles on intervention studies that aimed to improve data quality and use within routine health information systems in low- and middle-income countries, published in English from January 2008 to February 2020. We searched the literature in the databases Medline/PubMed, Web of Science, Embase, and Global Health. After a meticulous screening, we identified 20 articles on data quality and 16 on data use. We prepared and presented the results as a narrative.

### Results

Most of the studies were from Sub-Saharan Africa and designed as case studies. Interventions enhancing the quality of data targeted health facilities and staff within districts, and district health managers for improved data use. Combinations of technology enhancement along with capacity building activities, and data quality assessment and feedback system were found useful in improving data quality. Interventions facilitating data availability combined with technology enhancement increased the use of data for planning.

**Data Availability Statement:** All relevant data are within its supporting information files.

**Funding:** This work was supported by the Bill & Melinda Gates Foundation, grant number OPP1187448.

**Competing interests:** The authors have declared that no competing interests exist.

## Conclusion

The studies in this scoping review showed that a combination of interventions, addressing both behavioural and technical factors, improved data quality and use. Interventions addressing organisational factors were non-existent, but these factors were reported to pose challenges to the implementation and performance of reported interventions.

## Background

The health information system is one of the six building blocks of a health system and is designed to meet information needs within the health system. It generates information that is vital for planning, monitoring, and evaluating public health programs and interventions [1–3]. Decisions are made continuously at all levels of the health system. Information is generated that influences decisions from patient care to policy formation and implementation, thereby influencing health in the communities served by the health system [1]. The health information system generates information mainly from the routine health information system, which is composed of health service-based data, but also use population-based data from surveys, census, and vital event registrations [4]. The health information system performance is expressed as the quality of these data and their use for decision-making [1].

Thus, the quality of routine health information data is vital for the health system to function well and for policymakers to be able to evaluate the effects of health system efforts to improve the health of the population [2]. The quality of the routine health information system data has been enhanced across the globe [5, 6]. However, health systems in low- and middle-income countries are still suffering from a suboptimal quality and inadequate use of data generated by their routine health information systems [7–9]. The data quality issues are often expressed as incomplete registers [10, 11], lack of consistency between registers and reports [12–15], and low level of data accuracy [9]. Discrepancies between results from data generated in the routine health information system and population-based surveys are common [11].

Despite the increasing availability of health information, the use of such information for decision-making is still deficient in many low- and middle-income countries [16]. Studies from these settings show limited or inadequate use of data, especially of routinely generated data [17–20]. Studies at health facilities or based on interviews with health workers have frequently reported low use of such data for planning, despite these workers' engagement in the collection, aggregation, and generation of data reports to the next level in the health system [18, 20, 21]. A lack of trust in the quality of data may partly explain the limited use of data [17, 20].

The potential factors that affect routine health information system performance are classified as technical, behavioural, and organizational factors according to the performance of routine health information system (PRISM) framework [21]. Given the vital role of the routine health information system, there have been several interventions aiming to improve the quality and use of data targeting these potential factors [8, 22–26]. The design, methods, and scope of these interventions vary, making it difficult to conclude which interventions could be successful in what contexts.

The London School of Hygiene and Tropical Medicine implemented an intervention project called the Operational Research and Coaching for Analysts (ORCA) in collaboration with the Ethiopian Public Health Institute and the Ethiopian Ministry of Health. This project aimed to contribute to the improvement of data quality and use in the Ethiopian routine health

information system. This scoping review was part of these efforts and focused on the evaluation and synthesis of published results from interventions that aimed at improving data quality and use in routine health information systems in low- and middle-income countries. It answers the question: which interventions have successfully enhanced the quality and use of routine health information system data in low- and middle-income countries, and what contextual factors have influenced that outcome?

## Methods

### Protocol and registration

We developed a protocol before starting the work. However, we were not able to register at PROSPERO, since it had stopped registering systematic review at the time of request (S1 File).

### Study eligibility criteria

Peer-reviewed scientific journal papers were included in this review if the studies described or assessed interventions aimed at improving data quality or data use in the routine health information system. The studies should have been conducted in the health system in low- or middle-income countries and published in English in the period from January 2008 to February 2020. See Table 1 for inclusion and exclusion criteria.

We also included studies with any epidemiological design aiming at evaluating or describing a given intervention on the subject of interest, as well as qualitative evaluation studies.

Data quality is commonly defined based on its attributes or dimensions, and these dimensions vary with the different tools used. For instance, the WHO data quality review tool represents routine health information system data quality as data completeness, timeliness, consistency, and accuracy [27]. The data quality assessment tool prepared by the MEASURE evaluation group describe data quality as a primary dimension that consists of accuracy and reliability and the sub-dimensions precision, completeness, timeliness, integrity, and confidentiality [28]. Data use was defined as the use of routine health information system data for decision making at any level of the health system.

**Information source, search strategy, and selection.** To identify potentially relevant articles, we searched the following bibliographic databases from 2008 to 2018, with an additional search until February 2020: Embase, Medline/PubMed, Web of Science, and Global Health. Combinations of search terms were prepared based on the objective of the scoping review (SL) and further reviewed by an experienced information scientist (DW) to increase the certainty that the combination of the search terms answered our objectives. We reviewed reference lists

**Table 1. Summary of inclusion and exclusion criteria.**

| Inclusion criteria | Exclusion criteria |
|---|---|
| • Studies published in peer-reviewed scientific journals of any design meant to evaluate or describe interventions used to improve data quality or use of data from routine health information system | • Studies not focusing on routine health information system |
|  | • Grey literature |
|  | • Systematic reviews |
| • Studies published in English | • Language other than English |
| • Studies published from January 2008 to February 2020 | • Studies published before January 2008 |
| • Studies conducted in low- and middle-income countries according to the Development Assistance Committee (DAC)* list, 2018 | • Studies conducted in high-income countries |

*DAC: Development assistance committee

of published studies to check for saturation. The combination of search terms used is presented in S1A and S1B Box.

We exported the retrieved studies to the Zotero reference manager (Corporation for Digital Scholarship, Virginia, united states of America) and checked for duplicates. The first two reviewers (SL and AJ) screened titles and abstracts of the articles based on the eligibility criteria before reading the full articles in the Zotero reference manager. A third reviewer (CK) reviewed both titles and abstracts of the articles in case of disagreement between the first two reviewers. The decision made by the third reviewer was the final for inclusion or exclusion of the disputed studies. Finally, the full articles were reviewed in the same manner by the three reviewers.

## Data charting

We exported a list of the included studies into an Excel sheet. An Excel data abstraction tool was prepared, tested, and modified accordingly. The first two reviewers independently charted and compared the extracted data for any significant variation. The third reviewer further reviewed articles with a significant difference for a final decision (S1C Box).

## Synthesis

The presentation of the results followed the checklist for reporting of a scoping review "Preferred Reporting Items for Systematic Reviews and Meta-Analysis: extension for Scoping Reviews (PRISMA-ScR)" (28). S2 File. We described the search approach, and prepared tables summarising the characteristics of the included studies and their results. The analysis synthesised the type of interventions applied to improve data quality and use, as well as the effect of the reported interventions. The study also narrated potential opportunities and challenges of the reported interventions that may have influenced the outcome in low- and middle-income countries.

# Results

## Selection of sources of evidence

A total of 12 studies on the quality of data and 13 on data use were identified. Two studies were captured in both data quality and data use search processes. Three studies identified by the data use search process also dealt with data quality and were added to the data quality literature that finally included a total of 15 scientific papers. From the additional search for studies published January 2019- February 2020, we found one study that addressed data use alone, three studies on data quality and two on data quality and use. In total, we evaluated 20 and 16 studies on interventions on quality and use of data, respectively. The following flow diagrams (Figs 1 and 2) depicts the process of selection of literature and the search criteria.

## Characteristics of the literature

Out of the total of 20 studies on data quality, 14 were conducted in Africa, mostly in Sub-Saharan Africa [25, 29–31, 32–41], three in Asia (India, Pakistan, Sri Lanka) [42–44], and the remaining three in South America (Brazil, Mexico &Peru and Haiti) [45–47]. Out of the 16 articles included in the review of data use, ten were from Africa [25,30,36,37,39,48–52], mostly the Sub-Saharan region, five were from Asia (Philippines, Sri Lanka, India, and Iran) [43,44,53–55], and one from the Caribbean (Haiti) [56].

Most studies used a combination of interventions, such as the introduction of technology with training and supportive supervision. In general, the interventions on data quality fell into three major areas. The first and most common intervention on data quality was the use of

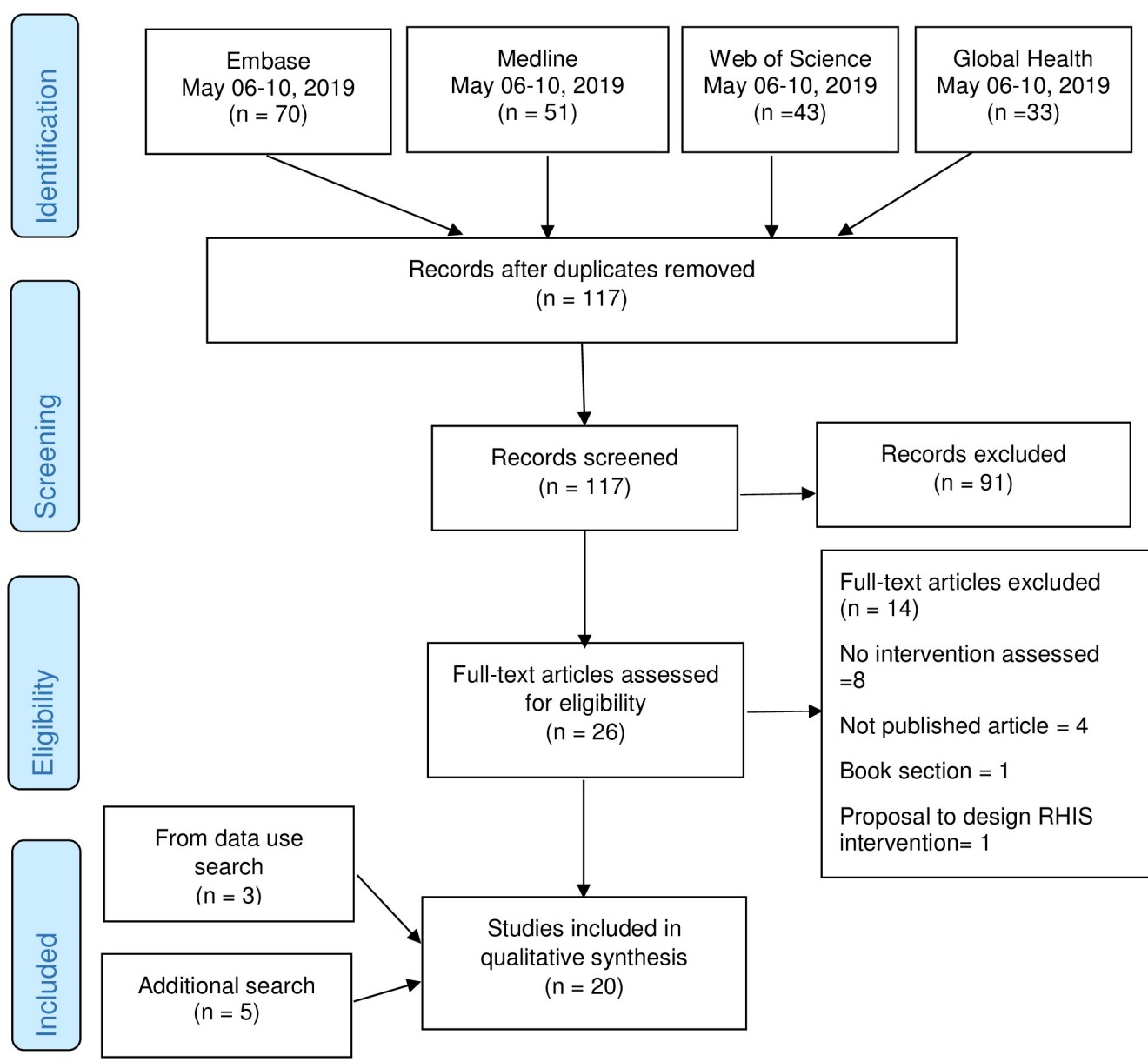

**Fig 1. The process and results of the literature search for data quality.**

technology, such as personal digital assistants (PDAs) for data entry, electronic form of data collection tools, such as electronic medical record (EMR), and electronic health management information system (EHMIS) to improve the data capturing at the health facility level. The second group of interventions was capacity building activities for personnel engaged in data collection, processing, and reporting at the health facility up to district level. The training was mainly on self-assessment and data quality management activities and how to use a framework for continuous improvements, such as the modified Plan-Do-Study-Act framework to systematically identify and act on data quality issues. The third group of interventions used evaluation tools to improve self-assessment and feedback systems in the routine health information system. These interventions encouraged the regular provision of feedback based on a systematic

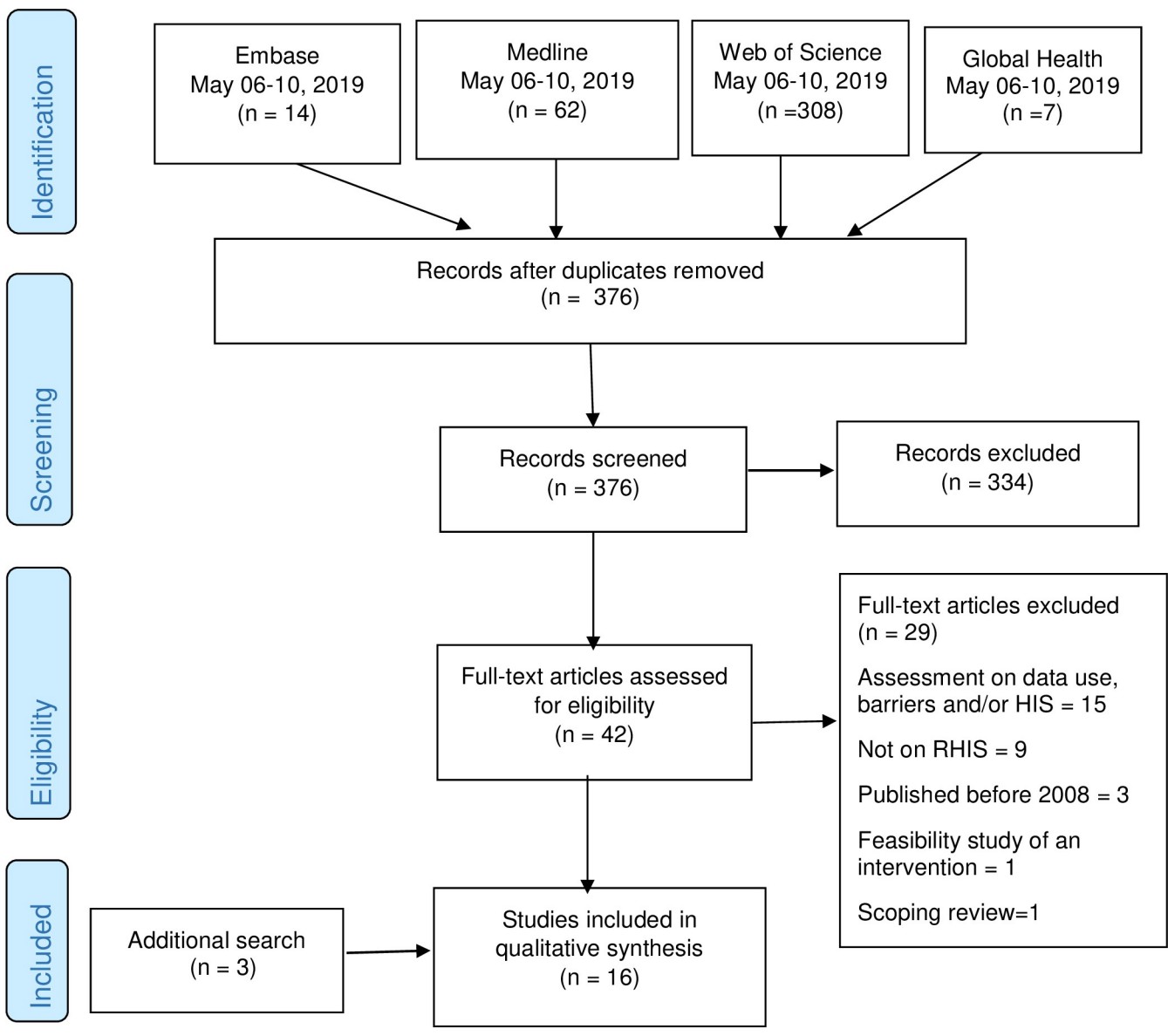

**Fig 2. The process and results of the literature search for data use.**

assessment of the data quality in the routine health information system. Table 2 summarises the data quality interventions.

Most of the studies on data use included combined interventions. These interventions also fell into three groups depending on the data use attributes the authors tried to improve or influence. The first group of interventions involved the use of tools or models facilitating decision-making. These tools or models should help decision-making by availing the necessary information in a logical and meaningful manner. The second group of interventions dealt with the use of technology to improve data quality and data use. These interventions also aimed to improve the availability of data for decision making. The third group of interventions was capacity building interventions, and only three articles fell under this category. Table 3 shows the details.

**Table 2. Interventions tested or adopted on specific health system populations aimed to improve data quality.**

| Author | Population | Interventions |
|---|---|---|
| 1. Use of technology | | |
| Ndira SP., 2008 [36] | Health system | Electronic health management information system compared to paper-based |
| Maokola W., 2011 [29] | Data collectors at health centres | Personal digital assistants |
| Monda J., 2012 [31] | Clinic health record system | Data integrity module: Plugin technology for auto-identification and correction of data capturing errors |
| Mutale W., 2013 [25] | Health system | Simplified information capturing system |
| Kaushik A., 2015 [43] | Secondary & tertiary care hospitals, primary care health centres and medical colleges | Enterprise architecture for e-health: web-based health management information system solution |
| Dombrowski JG., 2015 [45] | Routine health information system | Vertical health information system; live birth information system |
| Tuti T., 2016 [33] | Hospitals and clinicians treating children | Electronic tools. Research electronic data capture |
| Gimbel S., 2017 [34] | Health facilities and managers at the district level | Enhanced electronic medical record |
| Ismail S., 2017 [42] | Maternal and child health clinics | Standard-compliant data access model for maternal and child health data |
| Trumbo SP., 2018 [47] | Health facilities | Electronic immunization registers with multiple data-capturing systems |
| Rado, R., 2018 [35] | Health care structures in districts | Reinforced integrated disease surveillance response: Use of SMS for data transfer |
| Lazzerini, M., 2019 [44] | Hospitals | Development of an individual patient prospective database |
| 2. Capacity-building activities | | |
| Braa J., 2012 [30] | District health management team | Data use workshops for users |
| Mutali W., 2013 [25] | Health facility, district and province | Training of personnel managing data on data quality |
| Mpofu M., 2014 [32] | District monitoring and evaluation officers | Training and deploying monitoring and evaluation officers to manage data at the district |
| Wagenaar BH., 2017 [37] | Project principal investigators, implementers and Ministry of Health | Training of sub-national level managers on data analysis and output presentation approach |
| Gimbel S., 2017 [34] | Health facilities and managers at the district level | Continuous on-site mentorship |
| Njuguna, C., 2019 [40] | Health facilities | Development of customized guidelines for intergraded disease surveillance and response; training of focal person; |
| 3. Data quality assessments and feedback mechanisms | | |
| Mutale W., 2013 [25] | Health system: Districts and province, health centres, community health information system, dispensary and hospital | Routine provision of feedback on selected data quality issues; annual data quality assessment and feedback, community-level data assessment quarterly using Lots quality assurance. Quarterly data quality audit and automated data quality report based on logic error. Regular (monthly) review of reports and planned meetings between data generators and supervisors; district performance review and enhancement meetings. |
| Tuti T., 2016 [33] | Hospitals and clinicians treating children | Routine provision of feedback on data and mentorship; regular meetings three times a year |
| Puttkamer N., 2017 [46] | Site managers, clinicians, and data officers from hospitals and clinics using iSante | Automated data queries for extensive scale-site electronic medical record system (two strategies to establish priorities for data quality assessment) |
| Wagenaar B.H., 2017 [37] | Project principal investigators, implementers and Ministry of Health | Modified Plan-Do-Study-Act framework: facility-based data quality assessment, data review, and feedback meetings, data-driven action plan and follow-up of the plan |
| Gimbel S., 2017 [34] | Health facilities and managers at the district level | Developed data quality assessment for community health information systems, conducted quarterly data quality assessments, monthly data review meetings, and targeted supportive supervision |
| Njuguna, C., 2019 [40] | Health facilities | Monitoring, supervision and data quality assessment |

*(Continued)*

**Table 2.** (Continued)

| Author | Population | Interventions |
|---|---|---|
| O'Connor, EC., 2019 [39] | Community health workers and community leaders | Participatory community-based health information system |
| Yourkavitch, J., 2019 [41] | Community health workers | Data quality assessments to stimulate improvements to health management information systems |

* iSante: a multi-site electronic medical record implemented by the Haitian Ministry of Health.

## Design and target population

A good number of the studies lacked a clear description of the study methodology, i.e., their design and other methodological information. Four of the articles that dealt with data quality were case studies, which described an intervention, its implementation, and the observed

**Table 3. Interventions tested or adopted on specific health system populations aimed to improve data use.**

| Author | Population | Interventions |
|---|---|---|
| 1. Decision-making facilitation tools or models | | |
| La Vincente S., 2013 [53] | Rural province and urbanized city | Investment case approach: a decision support model that estimate the cost and impact of alternative interventions |
| Nutlay T., 2013 [49] | Health facilities in districts | District Health Profile: an excel based system linked with health facility data |
| Rajan D., 2014 [50] | Health System | Resource planning model based on the WHO integrated health care technology package |
| Kaushik A., 2015 [43] | Secondary and tertiary care hospitals, primary care health centres and medical colleges | Enterprise architecture for eHealth: Web-based health management information system solution that improves the availability of data for decision making |
| O'Connor, EC., 2019 [39] | Community health workers and community leaders | Participatory community-based health information system |
| 2. Use of technology to improve data quality and use | | |
| Gamur G., 2008 [48] | Primary care clinics | Feedback and Analytic Comparison tool: a form of a health information system at primary health care setting designed to facilitate data use for decision making |
| Ndira SP., 2008 [36] | Health system | Use of electronic health management information system to facilitate data quality and use |
| Matheson, AL., 2012 [56] | Hospitals where HIV care was provided | Use of electronic medical record system to facilitate data use |
| Mutale W., 2013 [25] | Clinics | Use of electronic data capture system to improve the quality of data improves data use at the health facility level |
| Mutale W., 2013 [25] | Community health information system workers at the community, dispensary, health centre and hospitals | Linking community-level health information data with routine health information system at facility and district |
| Hosseini M., 2014 [54] | Immunization records | Developing an immunization information system using service-oriented architecture and health level 7 to improve interoperability |
| Landis-Lewis Z., 2015 [51] | Anti-retroviral clinics and health care providers | Use of electronic medical record in anti-retroviral treatment clinics to improve the use of data for feedback |
| Nakibuuka, J., 2019 [38] | Health facilities | An Unstructured Supplementary Service Data-based health data reporting |
| Lazzerini, M., 2019 [44] | Hospitals | Development of a prospective Individual patient prospective database |
| 3. Capacity building to foster data use | | |
| Braa J., 2012 [30] | District Health Management team | Quarterly data use workshops for district health management team |
| Wagenaar BH., 2017 [37] | Project principal investigators, implementers, and Ministry of Health | Modified Plan-Do-Study-Act framework. Training of sub-national managers on data analysis and output presentation approach |
| Uneke JC., 2019 [52] | Policymakers | Policy information platform to improve access to information and thereby enhance the use of data for decision making and policy formation |

changes or improvement in data quality [25, 31, 33, 46]. Four reports had used a mixed-methods approach, where the authors had combined designs, including qualitative assessments [34, 35, 41, 47]. Six studies claimed to have an evaluation approach [30, 32, 35, 38, 39, 42] but without any well-articulated evaluation design, and two of the evaluation studies had employed a qualitative approach [30, 32]. Two papers reported cross-sectional studies [29, 45]. A majority of the articles on the use of data were case studies [25, 43, 48, 52, 55, 56, 58], two had used a mixed-methods approach [36, 54], three were qualitative studies [37, 50, 53], and two claimed evaluation designs that were not described [30, 36].

The target population for the data quality interventions was mainly health facilities at the district level and or their staff, including health care providers and data generators [29, 31, 33, 35, 38–42, 44]. Other studies targeted both the health facilities at the district level and the district management team [25, 34, 46, 47], and two reports solely focused on the district management team [30, 32]. The target population for data use interventions was similar to the data quality intervention studies. They focused on health facilities and their staff [48, 50, 53, 57, 58], or the health system in general, including its health management information system [36, 55, 56], or the management team, primarily at the district level [30, 37, 54], or a combination of the health facility and the district management team [25]. One study considered community health workers and the community health information system [25] (S1A and S1B Table).

## The outcome of the interventions as reported by the studies

Most studies used combinations of interventions. Fourteen of the 20 studies on data quality reported changes in data quality after the interventions [25, 29, 30, 32–34, 36–40, 44, 45, 47], and 11 of the 14 studies that reported change showed improvement in data quality [30, 32–34, 36–40, 44, 47]. For instance, the study by Mopuf and colleagues from Botswana, reported improved routine health information system data quality after training and deploying monitoring and evaluation officers at the district health office level [32]. Gimbel and colleagues [34] appreciated the use of combined interventions. Such examples were annual data quality assessments, provision of feedback to all districts using a summary data quality ranking tool, targeted supportive supervision for sites with weaker performance, in addition to monitoring and evaluation training. The latter study reported improved data availability from 84% to 99%, and a change in data consistency from 54% to 87% in Mozambique, one of their target countries [34]. Two of the studies reported a negative result (a decline in data quality performance) despite the interventions [29, 45]. A study in Tanzania introduced the use of personal digital assistants and trained data collectors at the health centre level, combined with monthly supervision and discussion on monthly generated reports without any positive effect on register completeness [29]. The second negative study established a live birth information system and compared its performance with an already available civil registry in Brazil [45]. The live birth information system performed poorly in coverage and completeness compared to the existing civil registry. The rest of the studies described the process and implementation of interventions that were meant to improve data quality without measuring the effect [31, 41–43, 46] (S1A Table).

The studies on data use also tested combinations of interventions. Eleven studies reported improvement in data use [25, 30, 37, 39, 43, 44, 48, 50, 52, 54, 57]. The study by Braa and colleagues in Zanzibar tested quarterly five-days use-of-data workshops for district health management teams included a peer review of performance on common data quality issues, after each team presenting their data. This study reported the adoption of simplified data capturing tools, increased use of data quality checks at facility levels and in districts, and improved alignment between plans, targets, and indicators [30]. Nutley and colleagues also tested a decision

support system labelled the district health profile in Kenya. This system was an Excel-based system with a set of standard questions on data quality. The authors reported improved data analysis, review, interpretation, and sharing of data, implying improved data use for decision making [49]. Wagenaar and colleagues also reported that most respondents reached the consensus that data were correct after implementing a modified Plan-Do-Study-Act framework in Zambia, Mozambique, and Rwanda [37]. This framework had steps such as identification of data quality problems, implementation of facility-based data quality assessment, training, feedback, data-driven action plan, supervision, and follow-up for action [37]. The rest of the studies reported a process of adoption of use-of-data interventions such as electronic medical record systems and their usefulness and acceptance by the targeted groups without assessing their effect on data use [55, 56, 58] (S1B Table).

### Reported barriers and positive attributes affecting interventions

A limited number of studies mentioned barriers or positive factors that could have influenced the interventions. No study tested these factors quantitatively in a statistical model. Besides, some of these factors reported were specific to each type of intervention assessed in the respective studies. We have summarized factors, which were crosscutting across the included papers. We broadly classified these factors as related to staff, resources, or infrastructure. Factors related to staff were lack of knowledge, skills, or training on a specific program or intervention, and inability to carry out the needed activities correctly or according to a set guideline [25, 35, 42, 45]. Some studies mentioned the lack of commitment or motivation to carry out a task or to adopt a new technology, which could affect either the implementation, the outcome, or both [25, 36, 45]. One study mentioned the lack of technical personnel as a barrier to reaching the intended result of the intervention [47]. Issues related to leadership, such as variation in the leadership quality and motivation of supervisors were reported as barriers [25]. At the same time, the presence of regular feedback facilitated a positive outcome of the intervention [35]. Lack of guidelines or protocols in the health system to guide the interventions [35], limited resources in general, interruption of funding [47], and inadequate technological infrastructure, such as shortage of computers and poor network connectivity, were reported as barriers [38, 42].

Similarly, some studies that dealt with the use of data reported on the barriers or favourable factors. Broadly, the elements summarised as issues related to the data quality, users, and resources. Poor data quality [48] and a limited availability of data [53, 55] were barriers to the implementation of data use interventions. Limited user acceptance of the intervention [51], limited capacity of users to access and use interventions [52], and users having little value for data or trust on the quality of data were also barriers [37, 50]. A persistent culture of non-use of data [37] was a barrier to the implementation as well as the outcome of these interventions. Some studies also reported resource constraints, such as access to computers and internet connectivity, as potential barriers to the success of interventions [50, 58].

### Discussion

In this scoping review, we identified 20 studies on interventions that targeted data quality and 16 that targeted data use. Most of these studies were from Sub-Saharan Africa, and most researchers had employed a case study approach. The main target groups for the interventions were district-level health facilities with staff or the public health system as such. Studies that dealt with the use of data for health planning also targeted district health managers. Interventions were frequently combined so that different aspects or attributes of data quality or use

were emphasised. Most of these combined interventions reported good progress and improved data quality and use in their respective target populations.

Data quality interventions that combined different capacity building activities, such as training and onsite mentoring, were reportedly effective. The combinations of capacity building activities with enhanced technical tools and data quality assessment combined with feedback systems were also useful. Two studies that did not use a combination of interventions reported persistent poor data quality in the routine health information system [29, 45]. Both studies focused only on technology enhancement, such as the use of electronic data capture, and the reports commented that technology enhancement alone might not bring the intended change. The latter study recommended routine investment in capacity building and regular data quality assessments.

Similarly, the use-of-data studies also stressed the importance of a combination of interventions. Those interventions facilitating data availability using standard tools combined with technology enhancement, such as the use of electronic data capture systems, were found useful in fostering data use. The technology enhancement interventions that aimed to improve data quality along with capacity building activities also reported positive changes in the use of data.

In light of the Performance of Routine Information System Management (PRISM) framework [21], the interventions in this scoping review targeted technical factors, which mainly dealt with the use of technology, and approaches that simplified activities in the routine health information system. The studies addressed factors, such as skills in performing data quality checks, problem-solving ability, and competence for the tasks in the health information system. Organizational factors were more rarely in focus, except for training and supervision. Overall, technical, organizational, and behavioural factors were partly addressed by the interventions. The relevance of untouched factors was undeniable, and they partially appear as listed barriers to the performance of the interventions and the implementation process by the studies. Some of these untouched factors were related to routine health information system governance or leadership, such as lack of funding, weak demand to use data, and lack of motivation concerning data quality and use.

The reviewed interventions did not involve higher-level managers and experts in the health system, as recommended in the PRISM framework. Although data are generated at lower levels of the health system, the contribution of experts and managers at all levels of the health system is crucial to realize better data quality and ensure continuous use of data at all levels. Managers and experts at higher levels in the health system are sources for many of the organizational factors, such as governance, finance, and promotion of information use that influence the performance of the routine health information system.

Data quality and data use interventions in high-income countries heavily relied on the use of technology. Examples are the use of electronic medical or health record systems to improve data quality [57–61] and approaches to enhance the interoperability of such data sources to enhance the availability of data and use [58, 60, 62, 63]. At the same time, these interventions were target-specific compared to the interventions in low- and middle-income countries, where targets were broader. Further, interventions in high-income countries primarily target health facilities. In contrast, studies in low- and middle-income countries addressed health service managers at the district level. Such variations could be explained by the difference in the relative importance of factors presented in the PRISM framework in these two different settings. Otherwise, both settings took advantage of the ever-changing technology advancement to improve the performance of their routine health information system.

This scoping review has strengths and limitations. We strictly followed the PRISMA-Scr guideline to maintain methodological rigor and transparency. Most of the scientific reports reviewed were case studies and lacked methodological rigour in evaluating a given

intervention. It was not possible to assess effect sizes when studies reported improvement in data quality or use. Changes in the outcome were reported as a change in percentage, or the authors used qualitative statements to show the presence of increment or improvement in the intended outcome, without providing test statistics. Thus, the results on the effectiveness of the interventions reported by the individual studies must be interpreted with caution. Furthermore, since this review included only published peer-reviewed articles, it may not be representative of all available literature in the field.

## Conclusion

The interventions reported in the reviewed studies targeted multiple dimensions of data quality and use. They called for combinations of interventions to enhance the performance of the routine health information system. There were positive effects when addressing both behavioural and technical factors in the routine health information system at the district health system level. There were few initiatives to target organizational factors that still pose a challenge to the performance of the system. Future routine health information system interventions should not only focus on technological solutions but target multiple factors at a time. Interventions should preferably also address organizational factors to influence the overall culture of data quality and use and also involve higher-level staff. Furthermore, intervention studies should employ an appropriate evaluation methodology that allows assessment of the effect of the intervention.

## Supporting information

**S1 Box.** A. Combinations of search terms formulated in Ovid for studies on data quality interventions. B. Combinations of search terms formulated in Ovid for studies on data use interventions. C. Contents of the data charting format used.
(DOCX)

**S1 Table.** A. Characteristics of studies on data quality interventions in low- and middle-income countries 2008–2020. B. Characteristics of studies on data use interventions in low- and middle-income countries 2008–2020.
(DOCX)

**S1 File. Protocol: Improving quality of routine health information system and data use in low- and middle-income countries: A scoping review.**
(DOCX)

**S2 File. Preferred Reporting Items for Systematic reviews and Meta-Analyses extension for Scoping Reviews (PRISMA-ScR) checklist.**
(DOCX)

## Acknowledgments

We are greatly indebted to our friend and colleague, Ms. Deepthi Wickremasinghe, who co-authored this article but sadly passed away before it was published. We also acknowledge the support from the library at London School of Hygiene and Tropical Medicine that availed some of the literature.

## Author Contributions

**Conceptualization:** Seblewengel Lemma, Carina Källestål.

**Data curation:** Seblewengel Lemma, Annika Janson, Carina Källestål.

**Methodology:** Seblewengel Lemma, Lars-Åke Persson, Deepthi Wickremasinghe, Carina Källestål.

**Writing – original draft:** Seblewengel Lemma.

**Writing – review & editing:** Seblewengel Lemma, Annika Janson, Lars-Åke Persson, Deepthi Wickremasinghe, Carina Källestål.

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
