## [Decision Letter · Decision Letter 0]

5 Aug 2020

PONE-D-20-12315

Improving quality and use of routine health information system data in low- and middle-income countries: A scoping review

PLOS ONE

Dear Dr. Lemma,

Thank you for submitting your manuscript to PLOS ONE. After careful consideration, we feel that it has merit but does not fully meet PLOS ONE’s publication criteria as it currently stands. Therefore, we invite you to submit a revised version of the manuscript that addresses the points raised during the review process.

We look forward to receiving your revised manuscript.

Kind regards,

Joel Msafiri Francis, MD, MS, PhD

Academic Editor

PLOS ONE

Journal Requirements:

2. We note in the Abstract on the online submission form you end it with a sentence to state "This work was supported by the Bill & Melinda Gates Foundation, Grant Number OPP1187448"

Please remove any funding-related text from the Abstract and let us know how you would like to update your Funding Statement. Currently, your Funding Statement reads as follows: "The authors received no specific funding for this work."Please include the updated Funding Statement in your cover letter. We will change the online submission form on your behalf.

3. Please note that in order to use the direct billing option the corresponding author must be affiliated with the chosen institute. Please either amend your manuscript or remove this option (via Edit Submission).

Reviewers' comments:

Reviewer's Responses to Questions

**Comments to the Author**

1. Is the manuscript technically sound, and do the data support the conclusions?

Reviewer #1: Yes

Reviewer #2: Yes

2. Has the statistical analysis been performed appropriately and rigorously? 

Reviewer #1: N/A

Reviewer #2: N/A

3. Have the authors made all data underlying the findings in their manuscript fully available?

Reviewer #1: Yes

Reviewer #2: Yes

4. Is the manuscript presented in an intelligible fashion and written in standard English?

Reviewer #1: Yes

Reviewer #2: Yes

5. Review Comments to the Author

Reviewer #1: PONE-D-20-12315

Improving quality and use of routine health information system data in low- and middleincome countries: A scoping review

Thank you for the opportunity to provide feedback on your research. This manuscript describes a well conducted review for interventioins to improve data quality and use in LMICs

I have provided some general thoughts for consideration.

Systematic vs scoping review

A scoping review is generally to map the literature as a whole on a particular topic in contrast to addressing a question which is the remit of the systematic review. So the scoping review is expected to be much broader in its inclusion and exclusion criteria and is expected to point out gaps in the availability of evidence rather than pointing out gaps in the studies per se.

This manuscript ends up looking like a systematic review rather than a scoping review due to its stringent inclusion criteria. A better question for the scoping review might have been ‘What are the characteristics and range of interventions used to improve data quality and data use for decision making in LMICs?’

Rather than looking at a wide variety of studies at how interventions have tried to improve the implementation of data quality and data use, this reviews focused more on the impact of individual studies and a critquee of thier evaluation designs. The makes it seem more like a systematic review and could very easily be presented as such. A consultative exercise with other domain experts would have been a great addition, if the plan is to persist with this as a scoping review.

Conisdering the narrowness of the scope and focus on specific class of intervetions (interventions to improve data use and quality), it may be better to rebrand it as a systematic review. In fact, the first paragraph of the method section seems to imply that this was conceptualised as a systematic review rather than a scoping one.

The restriction to peer reviewed and published studies maybe a limitation due to the fact that many such health systems strengthening activities are reported in grey literature and rare;y make it to peer reviewed journals.Again this makes the case for re branding it as a systematic review rather than a scoping review since a preference is made for strength of evidence in place of comprehensiveness ofinterventions.

Discussion

The discussion may be enhanced by the use of the PRISM framework early on in the text. in fact the PRISM framework could have been introduced (in the sysnthesis section of the methods) to help structure the results section as well. The first few paragraphs besides summing up seem to repeat many of the results already presented in the findings. yeah yeah

Minor points

1. There are a few places in the text where it would benefit from rewriting or rephrasing the sentences in simpler readable English. I've tried highlighting a few of them but may have missed some others.

2. I would actually be happy to see as a paragraph on the strengths of this review for example, the fact that rigorous and transparent methods have been used throughout from design to reporting, almost akin to a systematic review.

3. One of the stated limitations is a lack of experiences from high income countries. This may not be relevant for this scoping review since findings cannot be generalized to low income settings from studies in high income countries.

Reviewer #2: This manuscript provides a much-needed synthesis of the current evidence around interventions to improve routine data quality and use. The scoping review has been well-designed with methods presented in detail and according to PRISMA guidance, while the discussion has a good balance of depth without being overly long. I also appreciate the detailed supplementary materials. Overall, a really nice paper.

I have just a few of relatively minor clarifications and suggested edits.

1. On the cover sheet, I believe that the statement “This work was supported by BMGF, grant # …” should be moved from the abstract and included in the ‘financial disclosure’ section.

2. Abstract, line 41-42. I am not clear what is meant by “health facilities with staff at the district level”. Does this mean facilities within the district town only? Or simply that interventions targeting community health workers are excluded?

3. Background, line 61. I would argue that although continuous decision making at all levels of the health system is the aim, in practice, I think that some health systems are rather more top-down and no entirely decentralized.

4. Background lines 64-66. The definition of health information system used is surprisingly broad. I was not aware that it is standard practice for vital registration and survey data to be combined into a single system together with clinical and logistics data. Is this correct?

5. Background, line 79. Health information has not been previously defined. With a lot of different terms used the background section (routine health information, health information system, routine health information system, health service-based data, routine data, routine health information system, health information) there is a risk of confusing readers who are not already very familiar with these terms. You may want to consider including some definitions or minimizing the number of different terms used.

6. Methods, line 144-145. This final sentence of the paragraph repeats what has already been described and can be deleted.

7. Table 2. I would suggest including some kind of order in how each study is presented within each of the three sub-sections. This helps the reader to digest the information and see what is similar/different. For example, you could first list the interventions at ‘health system’ level, then those targeting referral facilities, facilities, and finally community-level.

8. Line 249. I think “assistance” should be “assistants”.

9. Discussion, lines 316-319. This statement is really important and should be highlighted (perhaps in the abstract or conclusion?) All too often researchers and implementors focus on technology solutions without exploring at other aspects of data quality – and your statement and the evidence from the review indicates that this approach is insufficient.

10. Discussion, line 346. The methods indicated that high-income countries are not included in the review.

11. Table S2A, page 5. Typos on country names for Sri Lanka and Sierra Leone. Same typos appear again on page 13.

6. PLOS authors have the option to publish the peer review history of their article (what does this mean?). If published, this will include your full peer review and any attached files.

Reviewer #1: No

Reviewer #2: No

---

## [Author Response · Author response to Decision Letter 0]

7 Sep 2020

Responses to Editor’s and reviewers’ comments

Comments from the Editor:

Authors’ response: We have checked and adhere to these requirements.

2. We note in the Abstract on the online submission form you end it with a sentence to state "This work was supported by the Bill & Melinda Gates Foundation, Grant Number OPP1187448"

Please remove any funding-related text from the Abstract and let us know how you would like to update your Funding Statement. Currently, your Funding Statement reads as follows: "The authors received no specific funding for this work."Please include the updated Funding Statement in your cover letter. We will change the online submission form on your behalf.

Authors’ response: We have followed this instruction. We want the funding statement to be:

This work was supported by the Bill & Melinda Gates Foundation, grant number OPP1187448.

3. Please note that in order to use the direct billing option the corresponding author must be affiliated with the chosen institute. Please either amend your manuscript or remove this option (via Edit Submission).

 Authors’ response: The publication fee will be covered by the Chronos mechanism ID:057AD34D-6CB3-4418-A2A2-A3D80F9BDF27

Comments from the Reviewers

Reviewer #1

Reviewer comment: Systematic vs scoping review

A scoping review is generally to map the literature as a whole on a particular topic in contrast to addressing a question which is the remit of the systematic review. So the scoping review is expected to be much broader in its inclusion and exclusion criteria and is expected to point out gaps in the availability of evidence rather than pointing out gaps in the studies per se.

This manuscript ends up looking like a systematic review rather than a scoping review due to its stringent inclusion criteria. A better question for the scoping review might have been ‘What are the characteristics and range of interventions used to improve data quality and data use for decision making in LMICs?’

Rather than looking at a wide variety of studies at how interventions have tried to improve the implementation of data quality and data use, this reviews focused more on the impact of individual studies and a critique of their evaluation designs. The makes it seem more like a systematic review and could very easily be presented as such. A consultative exercise with other domain experts would have been a great addition, if the plan is to persist with this as a scoping review.

Considering the narrowness of the scope and focus on specific class of interventions (interventions to improve data use and quality), it may be better to rebrand it as a systematic review. In fact, the first paragraph of the method section seems to imply that this was conceptualised as a systematic review rather than a scoping one. The restriction to peer reviewed and published studies maybe a limitation due to the fact that many such health systems strengthening activities are reported in grey literature and rarely make it to peer reviewed journals. Again this makes the case for re branding it as a systematic review rather than a scoping review since a preference is made for strength of evidence in place of comprehensiveness of interventions.

Authors’ response: We followed a strict procedure in producing this manuscript and used the PRISMA for a scoping review. This work was from the start conceptualized as a scoping review that intended to map interventions made in order to increase data quality and use in low- and middle-income countries. We cited the article by Munn et al. on the difference between systematic and scoping reviews (1) that states “The general purpose for conducting scoping reviews is to identify and map the available evidence” and “If the authors have a question addressing the feasibility, appropriateness, meaningfulness or effectiveness of a certain treatment or practice, then a systematic review is likely the most valid approach”. We strongly believe that our review fulfils the requirements for a scoping review. It describes the research area and points at some knowledge gaps (research on interventions on organizational factors to influence the overall culture of data quality and use) as well as limitations in methods used (appropriate evaluation methodology that allows measures of effects of interventions).

1. Munn, Z., Peters, M.D.J., Stern, C. et al. Systematic review or scoping review? Guidance for authors when choosing between a systematic or scoping review approach. BMC Med Res Methodol 18, 143 (2018). https://doi.org/10.1186/s12874-018-0611-x

We also acknowledge the potential benefit of also including non-peer reviewed grey literature. Our exclusion of such reports is acknowledged as a potential limitation on page 20, line 441 and 442.

Discussion

The discussion may be enhanced by the use of the PRISM framework early on in the text. in fact the PRISM framework could have been introduced (in the sysnthesis section of the methods) to help structure the results section as well. The first few paragraphs besides summing up seem to repeat many of the results already presented in the findings. yeah yeah

Authors’ response: The PRISM framework is crucial to this work. It is introduced early in the introduction section. We prefer not to use the PRISM framework to structure our results. The reason is that individual studies used multiple interventions at the same time. Using the PRISM framework for presenting the results would have complicated the text and reduced clarity. Therefore, we preferred a thematic presentation. However, we used the framework in the Discussion to interpret the results against this comprehensive and well-established framework. 

We started the Discussion section by summarizing the key findings. In our experience, this is conventional approach. 

Minor points

1. There are a few places in the text where it would benefit from rewriting or rephrasing the sentences in simpler readable English. I've tried highlighting a few of them but may have missed some others.

Authors’ response: We are grateful for the edits made by the reviewer; we have corrected those, now shown with track changes. We have carefully reviewed the text for similar language and style issues; also shown with track changes.

2. I would actually be happy to see as a paragraph on the strengths of this review for example, the fact that rigorous and transparent methods have been used throughout from design to reporting, almost akin to a systematic review.

Authors’ response: We do appreciate the comment. We have included this as a strength on page 20, line 434 and 435. 

3. One of the stated limitations is a lack of experiences from high income countries. This may not be relevant for this scoping review since findings cannot be generalized to low income settings from studies in high income countries.

Authors’ response: This sentence is now deleted. 

Reviewer #2 comment 

1. On the cover sheet, I believe that the statement “This work was supported by BMGF, grant # …” should be moved from the abstract and included in the ‘financial disclosure’ section.

Authors’ response: This istatement is removed and will be included in the appropriate place. 

2. Abstract, line 41-42. I am not clear what is meant by “health facilities with staff at the district level”. Does this mean facilities within the district town only? Or simply that interventions targeting community health workers are excluded?

Authors’ response: Sorry for the confusion. This is now rephrased as “Interventions enhancing the quality of data targeted health facilities and staff within districts, and district health managers for improved data use” and corrected on page 2, line 41. 

3. Background, line 61. I would argue that although continuous decision making at all levels of the health system is the aim, in practice, I think that some health systems are rather more top-down and no entirely decentralized.

Authors’ response: We agree that health systems frequently display top-down decision making. In this sentence we refer to the smaller day-to-day decision-making that is done to plan and execute daily activities. 

4. Background lines 64-66. The definition of health information system used is surprisingly broad. I was not aware that it is standard practice for vital registration and survey data to be combined into a single system together with clinical and logistics data. Is this correct? 

Authors’ response: According to the reference we cited (page 16) health information system include routine and non-routine sources of data, such as population-based surveys and vital registration data. We believe this is the generally accepted definition. 

5. Background, line 79. Health information has not been previously defined. With a lot of different terms used the background section (routine health information, health information system, routine health information system, health service-based data, routine data, routine health information system, health information) there is a risk of confusing readers who are not already very familiar with these terms. You may want to consider including some definitions or minimizing the number of different terms used.

Authors’ response: Thank you. We have minimized the number of different terms or synonyms used. Changes have been made on line 28, 68, 78, 137, 138, and 298. 

6. Methods, line 144-145. This final sentence of the paragraph repeats what has already been described and can be deleted.

Authors’ response: The last two lines describe the review process for full articles. We have rephrased for clarity. Please see the changes on page 8, line 172 and 173. 

7. Table 2. I would suggest including some kind of order in how each study is presented within each of the three sub-sections. This helps the reader to digest the information and see what is similar/different. For example, you could first list the interventions at ‘health system’ level, then those targeting referral facilities, facilities, and finally community-level.

Authors’ response: Thank you for the suggestion. We have now arranged the studies chronologically. To be consistent, we have also arranged the content in Table 3 in the same manner. See Table 2 & 3. 

8. Line 249. I think “assistance” should be “assistants”.

Authors’ response: This is now corrected on page 15, line 307. 

9. Discussion, lines 316-319. This statement is really important and should be highlighted (perhaps in the abstract or conclusion?) All too often researchers and implementors focus on technology solutions without exploring at other aspects of data quality – and your statement and the evidence from the review indicates that this approach is insufficient.

Authors’ response: Thank you. We have now rephrased this in the concluding sentences, on page 21, line 459 &460.

10. Discussion, line 346. The methods indicated that high-income countries are not included in the review. 

Authors’ response: Thank you for this comment. We would like to keep this. The reason is that we referred to studies in high-income countries to provide some contrast to the results from the low- and middle-income countries. 

11. Table S2A, page 5. Typos on country names for Sri Lanka and Sierra Leone. Same typos appear again on page 13.

Authors’ response: All typos are now corrected in the revised version see page 6 and 13.

---

## [Decision Letter · Decision Letter 1]

11 Sep 2020

Improving quality and use of routine health information system data in low- and middle-income countries: A scoping review

PONE-D-20-12315R1

Dear Dr. Lemma,

We’re pleased to inform you that your manuscript has been judged scientifically suitable for publication and will be formally accepted for publication once it meets all outstanding technical requirements.

Kind regards,

Joel Msafiri Francis, MD, MS, PhD

Academic Editor

PLOS ONE

Additional Editor Comments (optional):

Reviewers' comments:

Reviewer's Responses to Questions

**Comments to the Author**

1. If the authors have adequately addressed your comments raised in a previous round of review and you feel that this manuscript is now acceptable for publication, you may indicate that here to bypass the “Comments to the Author” section, enter your conflict of interest statement in the “Confidential to Editor” section, and submit your "Accept" recommendation.

Reviewer #1: All comments have been addressed

Reviewer #2: All comments have been addressed

2. Is the manuscript technically sound, and do the data support the conclusions?

Reviewer #1: Yes

Reviewer #2: (No Response)

3. Has the statistical analysis been performed appropriately and rigorously? 

Reviewer #1: N/A

Reviewer #2: (No Response)

4. Have the authors made all data underlying the findings in their manuscript fully available?

Reviewer #1: Yes

Reviewer #2: (No Response)

5. Is the manuscript presented in an intelligible fashion and written in standard English?

Reviewer #1: Yes

Reviewer #2: (No Response)

6. Review Comments to the Author

Reviewer #1: Thank you for responding to my feedback. This is a very useful study and a well written manuscript.

My comment about repeating the results in the discussion may have led to some confusion about the intention. I agree it is the usual practice to summarize the results as part of the discussion. My point was that it could have been kept shorter to keep the manuscript tighter and sustain reader interest. For eg, line 311- 316, you describe the findings of two studies which would have been more appropriate for the results rather than the discussion. The pertinent point for the discussion would have been that digital enhancements alone do not bring about improvement in data quality.

Reviewer #2: (No Response)

7. PLOS authors have the option to publish the peer review history of their article (what does this mean?). If published, this will include your full peer review and any attached files.

Reviewer #1: No

Reviewer #2: No

---

## [Editor Report · Acceptance letter]

16 Sep 2020

PONE-D-20-12315R1 

Improving quality and use of routine health information system data in low- and middle-income countries: A scoping review 

Dear Dr. Lemma:

I'm pleased to inform you that your manuscript has been deemed suitable for publication in PLOS ONE. Congratulations! Your manuscript is now with our production department. 

Kind regards, 

on behalf of

Dr. Joel Msafiri Francis 

Academic Editor

PLOS ONE